# Tolerability and pharmacokinetic evaluation of inhaled dry powder hydroxychloroquine in healthy volunteers

Y. A. de Reus[1], P. Hagedoorn[2], M. G. G. Sturkenboom[3], F. Grasmeijer[2,4], M. S. Bolhuis[3], I. Sibum[2], H. A. M. Kerstjens[1], H. W. Frijlink[2], O. W. Akkerman[1,5]*

1 Department of Pulmonary Diseases and Tuberculosis, University of Groningen, University Medical Center Groningen, Groningen, The Netherlands, 2 Department of Pharmaceutical Technology and Biopharmacy, University of Groningen, Groningen, The Netherlands, 3 Department of Clinical Pharmacy and Pharmacology, University of Groningen, University Medical Center Groningen, Groningen, The Netherlands, 4 PureIMS B.V., Roden, The Netherlands, 5 TB Center Beatrixoord, University of Groningen, University Medical Center Groningen, Groningen, The Netherlands

* o.w.akkerman@umcg.nl

**Data Availability Statement:** All relevant data are within the paper and its Supporting information files.

## Abstract

### Rationale

Inhaled antimicrobials enable high local concentrations where needed and, compared to orally administration, greatly reduce the potential for systemic side effects. In SARS-CoV-2 infections, hydroxychloroquine sulphate (HCQ) administered as dry powder via inhalation could be safer than oral HCQ allowing higher and therefore more effective pulmonary concentrations without dose limiting toxic effects.

### Objectives

To assess the local tolerability, safety and pharmacokinetic parameters of HCQ inhalations in single ascending doses of 5, 10 and 20 mg using the Cyclops dry powder inhaler.

### Methods

Twelve healthy volunteers were included in the study. Local tolerability and safety were assessed by pulmonary function tests, electrocardiogram and recording adverse events. To estimate systemic exposure, serum samples were collected before and 0.5, 2 and 3.5 h after inhalation.

### Results and discussion

Dry powder HCQ inhalations were well tolerated by the participants, except for transient bitter taste in all participants and minor coughing irritation. There was no significant change in QTc-interval or drop in $FEV_1$ post inhalation. The serum HCQ concentration remained below 10 µg/L in all samples.

**Funding:** The production, labelling and distribution of the HCQ dry powder Cyclops inhalers was provided free of charge by PureIMS. The company provided support in the form of salaries for authors [FG], but did not have any additional role in the study design, data collection and analysis, decision to publish, or preparation of the manuscript. All other funding was supplied by the institutions of the participating authors at University of Groningen and University Medical Center Groningen. The specific roles of these authors are articulated in the 'author contributions' section'. We declared the use of a patented inhaler called the Cyclops with number WO2015/187025. This does not alter our adherence to PLOS ONE policies on sharing data and materials.

**Competing interests:** I have read the journal's policy and the authors of this manuscript have the following competing interests: The employer Prof. Dr. H.W. Frijlink and P. Hagedoorn has a license agreement with PureIMS on the Cyclops patent (WO2015/187025) and receives royalty payments over the sales of the Novolizer®, Genuair®/Pressair® and Cyclops dry powder inhalers. In addition, Prof. Dr. H.W. Frijlink and Paul Hagedoorn have a patent WO2015NL50413 20150605 licensed. Floris Grasmeijer is employed by PureIMS, the manufacturer of the Cyclops inhaler. This does not alter the authors' adherence to PLOS ONE policies on sharing data and materials. The other authors have no conflicts of interest to declare.'

## Conclusion

Single doses of inhaled dry powder HCQ up to 20 mg are safe and well tolerated. Our data support that further studies with inhaled HCQ dry powder to evaluate pulmonary pharmacokinetics and efficacy are warranted.

## Introduction

In late December 2019 an outbreak of the novel coronavirus, Severe Acute Respiratory Syndrome Coronavirus 2 (SARS-CoV-2), started in Wuhan, China, and caused the spread of corona virus disease 2019 (COVID-19) [1]. The WHO declared the epidemic of COVID-19 a pandemic on March 12th, 2020 [2]. The virus is still rapidly spreading with over 542 million cases and 6.3 million deaths reported worldwide by the end of June 2022 [3].

Since the emergence of SARS-CoV-2 many treatment options have been studied. Current treatment for patients needing hospitalization with oxygen therapy consist of dexamethasone with addition of IL-6 inhibitors in patients with high inflammation and respiratory deterioration necessitating high oxygen supply [4]. Monoclonal antibodies have a role in reducing hospitalizing rates and mortality in high risk patients without endogenous antibody production but are of less value with the predominance of omicron-variants that are not sufficiently neutralized by these antibodies [5]. The latest recommendation is the administration of nirmatrelvir/ritanovir within 5 days of symptom development to patients with non-severe disease with the highest risk of hospitalization because of a 89% reduction in risk of progressing tot severe COVID-19 compared to placebo [4, 6]. In addition to an effective treatment, there is also the need to be able to prevent and decrease transmission in the general population and moreover in healthcare workers or other high-risk groups. Next to the current vaccination strategies, alternatives should be investigated for both treatment in early disease and prevention of transmission. Repositioning old drugs for use as antiviral or anti-inflammatory treatment is an interesting strategy because the safety profile, side effects, posology and drug interactions are already known which can speed up the trial program duration considerably.

Among those drugs is hydroxychloroquine sulphate (HCQ), which is mostly used in rheumatologic conditions because of its immune-modulatory effects [7, 8]. HCQ has proven to be effective in *in vitro* Vero cell systems infected with SARS-CoV-2 in two separate Chinese studies [9, 10]. Angiotensin converting enzyme 2 (ACE2) receptor expressing cells play an important role in the pathogenesis of SARS-CoV-2 infection, as the virus uses this receptor to enter the cell [9, 11–13]. HCQ impairs the terminal glycosylation of ACE2 and thereby inhibits cell-binding and entry of the virus into the cell [14–16]. Furthermore, HCQ also blocks transport of SARS-CoV-2 from early endosomes to endolysosomes, a requirement to release the viral genome [9, 10, 17]. Finally, HCQ has anti-inflammatory properties as it influences the generation of pro-inflammatory cytokines and endosomal inhibition of toll-like receptors, which have a major role in innate immune response [9, 18–21]. Based on the *in vitro* findings, oral HCQ was used abundantly worldwide in the beginning of the COVID-19 pandemic, both off label and in clinical trials. Some observational studies showed clinical benefit and antiviral effects [22–26] while others did not [27, 28] or were inconclusive [29, 30]. Currently, both the FDA and EMA advice against the off-label use of oral HCQ based on the large clinical RECOVERY trial that showed no beneficial effects on 28-day mortality [31]. The prospective European DISCOVERY trial and WHO SOLIDARITY trial have discontinued the oral HCQ treatment because of a lack of effect on mortality arms as well. The failing treatment with oral HCQ may be explained by insufficient concentrations in alveolar epithelial cells due to its

large volume of distribution of 5500 liter [32]. Raising the oral dose is not an option, since this is limited by adverse or even toxic effects, including the risk of cardiovascular toxicity (QTc prolongation).

Pulmonary administration of HCQ can be the solution to reach high local pulmonary concentrations without systemic toxicity [33, 34]. For this purpose, we developed a dry powder formulation of HCQ suitable for inhalation using the Cyclops dry powder inhaler. The Cyclops is a high dose disposable inhaler that enables effective dispersion of up to 50 mg of active ingredient. It emits a high fraction of the total dose in the respirable size range of 1 to 5 μm. Furthermore, its medium to high resistance to airflow limits the inhalation flow rate [35]. These factors combined prevent substantial drug deposition in the oropharynx and enable the deposition of the drug in the small airways and alveoli. The aim of this study was to assess local tolerability and safety of increasing doses of dry powder HCQ administered using the Cyclops.

## Methods

### Study design

This study was an open-label phase 1a single ascending dose study with twelve healthy volunteers, aged $\geq$ 18 years. Participants were recruited by advertisement between July and September 2020. The study was performed at the University Medical Center Groningen (UMCG) location Beatrixoord (Haren, the Netherlands) between September 29[th] 2020 and October 16[th] 2020. Participants were administered HCQ dry powder per inhalation using the Cyclops in single doses starting with 5 mg and ascending to 10 mg and 20 mg with a wash-out period between doses of at least 4 days. In- and exclusion criteria are listed in Table 1. G6PD-deficiency and pregnancy were excluded before start of the study if applicable. The study was approved by the hospital medical ethical review committee (UMCG, Groningen, The Netherlands, METc number 2020.168). The study was performed according to the Helsinki declaration (Fortaleza, Brazil, 2013) and was registered at clinicaltrials.gov (NCT04497519). Written informed consent was obtained from the participating subjects.

### Study drug

The dry powder formulation of HCQ was developed at the department of Pharmaceutical Technology and Biopharmacy of the University of Groningen, Groningen, the Netherlands. HCQ sulphate was obtained from Ofipharma B.V. (Ter Apel, the Netherlands). The HCQ Cyclops was produced by PureIMS B.V. (Roden, the Netherlands). For this, hydroxychloroquine sulphate was comicronized by air jet milling with 4% L-leucine to improve formulation

**Table 1. In- and exclusion criteria.**

| Inclusion criteria |
| --- |
| Healthy volunteer |
| Age 18–65 years |
| Obtained written informed consent |

| Exclusion criteria |
| --- |
| Pregnancy or breastfeeding |
| Contra-indication to (hydroxy)chloroquine or quinine (allergic reaction, prolonged QTc-interval ($>$ 450 msec), long-QT syndrome (LQTS), retinopathy, epilepsia, myasthenia gravis, G6PD-deficiency) |
| Concurrent use of ciclosporin, digoxin, ritonavir, tamoxifen or tranylcypromine. |
| Concurrent use of high risk QTc prolongating drugs (amiodarone, erythromycin (daily dose $>$ 1000 mg) or sotalol) |
| COVID-19 like symptoms, such as fever, cough, or sore throat; only by history taking. |

dispersion. Each Cyclops contained a nominal dose of 5 mg or 10 mg HCQ; the 20 mg dose was administered as 2 successive inhalations with 10 mg of HCQ. The doses were comparable with known doses of inhaled nebulized HCQ and thus expected to be safe [34, 36]. Furthermore, to enhance dose emission from the inhaler, 5 mg of coarse lactose was added to the dose compartment. At 4 kPa, 85% of the nominal dose was emitted from Cyclops with a fine particle fraction < 5 μm of 74% (i.e. 63% of the nominal dose), as determined by laser diffraction analysis.

## Objectives and procedures

The primary objective was to assess local tolerability and safety. Local tolerability was assessed by spirometry combined with active questioning about adverse events experienced by the participants. A drop of the forced expiratory volume in the first second (FEV$_1$) of 15% or more after inhalation of HCQ compared to baseline FEV$_1$ was considered clinically significant and critical to decide on proceeding with the next ascending dose. If a drop in FEV$_1$ occurs this is expected soon after inhalation as shown in other studies with inhaled antimicrobials [37] and therefore spirometry was performed before inhalation (baseline), 35 and 95 minutes after inhalation of HCQ according to the ATS/ERS guidelines [38]. Adverse events were continuously assessed during the study day. Cough for more than one hour or any other reported adverse event that made either the physician or the participant decide to stop participation was considered critical to decide on proceeding with the next ascending dose. Electrocardiograms (ECGs) were performed to assess the QTc interval as safety parameter. An ECG was obtained at the screening visit, before inhalation of the first dose and at the end of each study day, approximately 3.5 hours after each HCQ inhalation. An observed QTc interval of more than 500 ms was also considered critical on proceeding with the next ascending dose. All tolerability and safety endpoints were discussed with a Data Safety Monitoring Board (DSMB) after all twelve participants completed a dose step before proceeding to the next ascending dose.

The secondary objectives were to assess systemic exposure of HCQ and measurement of inspiratory flow parameters. To determine the systemic exposure, blood samples were collected from an intravenous indwelling cannula just before inhalation (predose), 30 minutes, 2 and 3.5 hours after inhalation of HCQ. The timepoints were chosen based on the absorption rate of HCQ after oral administration with a time to maximum concentration of 2–4.5 hours [39], because data after inhalation were not available at that time. The samples were analyzed using a validated liquid chromatography tandem mass spectrometry (LC-MS/MS) method at the laboratory of the department of Clinical Pharmacy and Pharmacology of the UMC Groningen (ISO15189:2012 (M170) certified). The limit of quantification of the method was 10 μg/L. The delivered dose was determined by subtracting the powder residue inside the Cyclops after inhalation from the exact weighed dose pre inhalation. The powder residues in the Cyclops were dissolved in demineralized water and the solutions were analyzed with a Thermo Scientific spectrophotometer (Genesys 150 UV–VIS, The Netherlands) at a wavelength of 236.0 nm.

Prior to inhalation of the study drug, study participants received inhalation instructions followed by training regarding handling of the device and performing a correct inhalation maneuver. Instruction was done using an empty Cyclops connected to a laptop, with in-house developed software application (labVIEW, National Instruments, Groningen, the Netherlands) for recording and processing of flow curves generated through the device. When a series of consistent flow curves meeting the criteria for good inhaler performance was obtained during training, a Cyclops containing HCQ was handed to the participant. Inspiratory flow parameters were recorded during each drug administration.

## Data management and statistics

Study data were collected and managed using Research Electronic Data Capture (REDCap) [40, 41]. Statistical analysis was performed with SPSS version 23. Data from all subjects who received at least one dose of the study drug were included in analyses of safety. Data were summarized using descriptive statistics. At each visit and timepoint, testing for differences in pre- to post-dose changes in $FEV_1$ (liters) and QTc interval were performed using the paired T-test or Wilcoxon rank sum test. P values below 0.05 were considered statistically significant.

## Results

Twelve participants were screened and enrolled, and all completed the study (Fig 1). In Table 2, patient characteristics are presented. All participants had normal baseline QTc interval and G6PD-deficiency was excluded in all participants. At the screenings visit pregnancy was excluded in female participants.

There were no serious adverse events observed during or after the study (Table 3). None of the participants had cough longer than the pre-defined safety period of one hour. Minor complaints of cough were reported four times, each in a different participant, directly after inhalation of HCQ; two times after a dose of 10 mg HCQ and two times after a dose of 20 mg HCQ. Complaints varied from experience of an itchy or tickling sensation in the throat to a single or a few observed coughs that were self-limiting. Two participants reported minimal dyspnea which disappeared after coughing or spontaneously within 4 minutes after inhalation. All participants mentioned bitter taste after inhalation. In most participants, this lasted for 5–10 minutes, but one participant reported the bitter taste for 2 hours. Participants were advised to rinse their mouth with water or eat something after the dose was administered with satisfying effect. Two participants had complaints of slight nausea relating to the bitter taste; one for 5 minutes after a dose of 5 mg HCQ and one for 70 minutes after a dose of 10 mg HCQ. Sore throat was reported by three participants. Two participants developed symptoms of upper respiratory tract infection with sore throat in the following days and were assessed by PCR nasopharyngeal swab for respiratory viruses of which one tested positive for SARS-CoV-2 two days after the last study visit. One participant experienced some hoarseness, which disappeared after drinking some water. Two other adverse events were recorded. One participant with an intrauterine contraceptive device in situ experienced minor spotting two days after inhalation of HCQ after both the 5 mg and 10 mg dose. This is not mentioned as a known side-effect of oral HCQ [42]. One participant mentioned dry eyes once at a dose of 20 mg, which seemed to be related to the dry hospital environment and not to administration of the study drug.

None of the participants showed a significant drop in $FEV_1$ ($\geq$15%) at any time point after HCQ administration (Table 4). A maximum drop of 7.51% was observed 95 minutes after inhalation of 5 mg HCQ. This participant mentioned a sort of burning sensation at the chest at that moment as well but no dyspnea. He recognized this burning sensation as similar while running.

None of the participants had a QTc prolongation $\geq$ 500, nor $\geq$ 450 ms. Mean QTc interval was 412 ms (range 384–441) at baseline. This did not change significantly 3.5 hours after inhalation of 5 mg, 10 mg or 20 mg HCQ, respectively (Table 5).

In all participants, the serum HCQ concentrations sampled predose, 30 minutes, 2 and 3.5 hours, were below the detection level of 10 μg/L, irrespective of dose. The mean delivered dose was 3.16 mg (63%) after administration of 5 mg HCQ, 6.95 mg (70%) after administration of 10 mg HCQ and 14.86 mg (75%) after administration of 20 mg HCQ. Based on the recorded inspiratory flow parameters, all participants correctly performed the inhalation maneuvers.

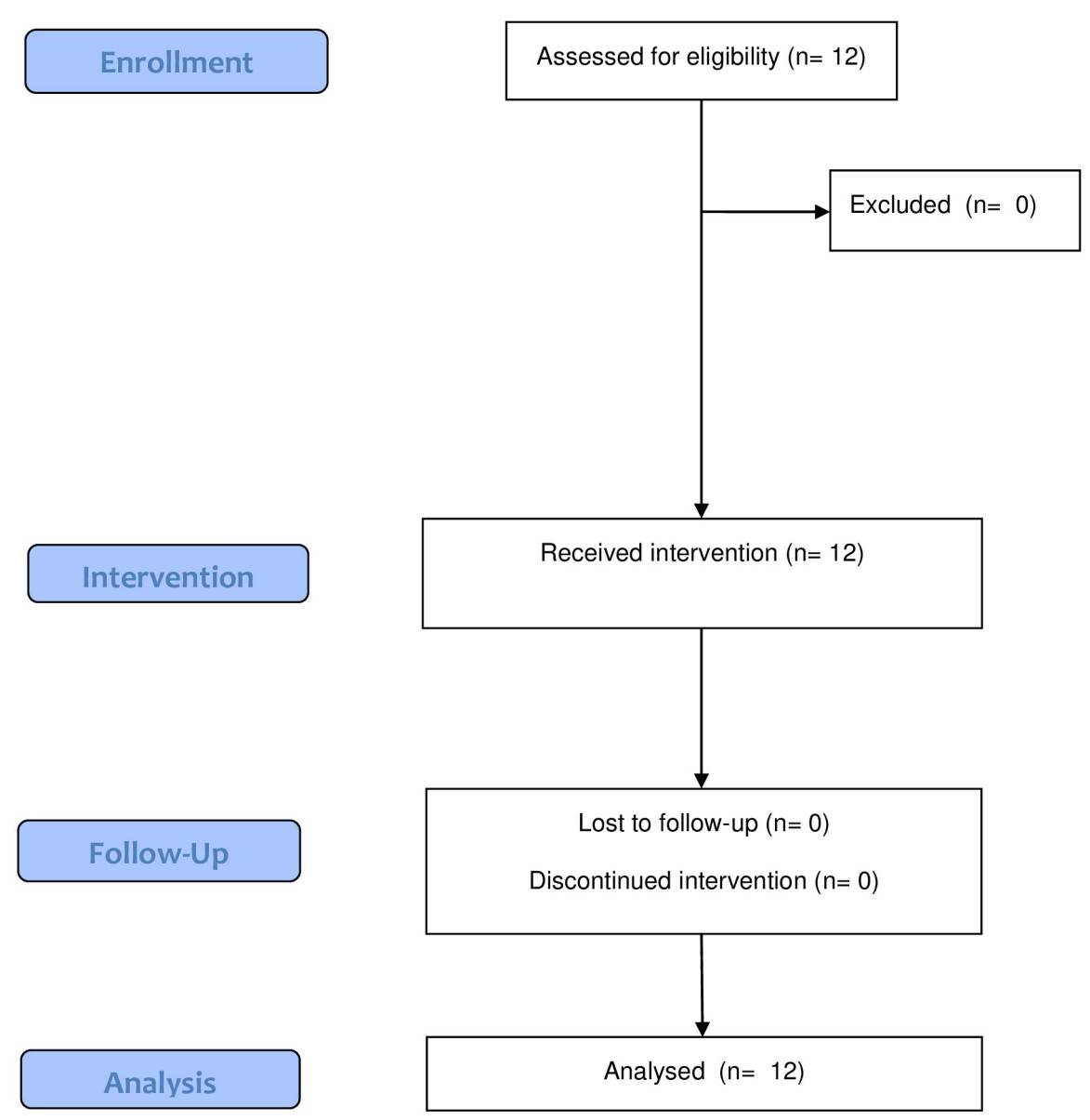

**Fig 1. Flow diagram of participants enrolled and analyzed in this study.**

**Table 2. Participant characteristics.**

|  | *N (%) or Mean (range)* |
| --- | --- |
| *Sex (male / female), N (%)* | *9 (75) / 3 (25)* |
| *Age (years)* | *30 (20–53)* |
| *FEV$_1$ predicted (%)* | *101 (90–112)* |
| *BMI (kg/m$^2$)* | *26.2 (21.8–38.5)* |
| *Non-smoking / Ex-smoking / Current smoker, N (%)* | *4 (33) /4 (33)/ 4 (33)* |

**Table 3. Reported adverse events out of 36 HCQ administrations by inhalation in 12 participants.**

| Adverse events | 5 mg | 10mg | 20 mg | Percentage of total administrations |
|---|---|---|---|---|
| | N (%) | N (%) | N (%) | N (%) |
| Cough | 0 (0) | 2 (17) | 2 (17) | 11 (31) |
| Dyspnea | 0 (0) | 1 (8) | 1 (8) | 6 (17) |
| Bitter taste | 12 (100) | 12 (100) | 12 (100) | 100 (100) |
| Nausea | 1 (8) | 1 (8) | 0 (0) | 6 (17) |
| Sore throat | 1 (8) | 1 (8) | 1 (8) | 8 (22) |
| Hoarseness | 0 (0) | 1 (8) | 0 (0) | 3 (8) |
| Spotting | 1 (8) | 1 (8) | 0 (0) | 6 (17) |
| Dry eyes | 0 (0) | 0 (0) | 1 (8) | 3 (8) |

## Discussion

Pulmonary administration of HCQ for early COVID-19 treatment or prevention in post-exposed individuals might be an alternative to oral administration of HCQ, as oral HCQ is complicated by adverse events and has not shown clinical relevance in the treatment of COVID-19 contrary to the expectation based on the mechanism of action and promising in vitro results [9, 10, 31]. Hypothetically, HCQ concentration in the lungs after oral dosing might be too low to exert an effect and with HCQ inhalations higher pulmonary concentrations can be reached compared to oral HCQ while using much lower doses and exerting lower systemic exposure. The proposition of inhaled HCQ might have a role in decreasing disease severity and transmission. This phase 1 study with three different single doses up to 20 mg of inhaled HCQ showed good local tolerability and safety without significant systemic side effects.

Coughing was reported four times out of a total of 36 administered doses (11%) in four different participants and was very mild in severity. Coughing is often reported after inhalation of antimicrobials by both wet nebulization and dry powder inhalation, with data mainly available from cystic fibrosis patients treated with colistin or tobramycin. In general, cough is more frequently reported after dry powder inhalation (ranging from 75%–90%) than after nebulization (ranging from 31%–78%) in these patients, although some studies found no differences [43–45]. The same might be true for HCQ, since cough was not reported by asthma patients in phase 1 and 2 studies with aerosolized HCQ [34]. The probable trigger for both cough and bitter taste is deposition of the drug in the oropharynx. This could be expected since HCQ is a quinoline known for its extreme bitter taste (249 on a bitter scale compared to caffeine at 46) [46]. That taste masking was not a problem in the studies with aerosolized HQC might be due to the use of another device and a low dosage volume of only 50 μl [34]. However, the majority of participants in our study reported that the taste was not disturbing since the participants were warned beforehand, it was minor, and it disappeared within a few minutes after HCQ inhalation or even faster when rinsing the mouth with water or eating something directly after inhalation.

**Table 4. Change in FEV1 post inhalation compared to baseline in %: Mean and range.**

| Dose | 35 min post inhalation | 95 min post inhalation |
|---|---|---|
| 5 mg HCQ | -0.98 (-5.44 –+1.90) | -1.21 (-7.51 –+2.44) |
| 10 mg HCQ | -0.21 (-7.30 –+3.79) | 0.21 (-5.31 –+4.66) |
| 20 mg HCQ | -0.98 (-5.44 –+1.90) | -1.21 (-7.51 –+2.44) |

 

**Table 5. QTc time at baseline and 3,5 hours after inhalation.**

|  | Qtc time in ms Mean (range) |
|---|---|
| **Baseline** | 412 (384–441) |
| **Post 5 mg HCQ** | 407 (383–439) |
| **Post 10 mg HCQ** | 414 (392–447) |
| **Post 20 mg HCQ** | 409 (381–423) |

Bronchus obstruction was not a problem after inhalation of HCQ dry powder; none of the twelve participants experienced a drop in $FEV_1$ of 15% or more. The maximum drop was 7.5% compared to baseline and this was not accompanied by dyspnea. One participant did mention a light burning sensation at the chest at that moment, something he also experienced while running. Although not formerly diagnosed, this participant might suffer from a mild exercise induced asthma.

We added ECG assessment as a safety parameter, although (high) systemic concentrations were not expected after local administration in a dose that is just a fraction of the oral dose. HCQ inhalations did not lead to prolongation of the QTc interval $\geq$ 450ms in any participant. The concerns about cardiotoxicity of oral HCQ, a well-known drug and generally considered to be safe and well tolerated, has arisen by extrapolating long-term risks of myocardial damage with chronic dosing to short-term exposures, thereby overestimating the risk of ventricular arrhythmias [32]. Our study indicates that systemic exposure after inhalation is very low, which is a positive result regarding the risk of systemic toxicity. Also, the concerns for any other possible systemic adverse event should be tempered because of these results.

A limitation of our study is the small sample size which might not be representative of a larger population, since we only studied twelve participants and predominantly men. It is difficult to make a specific calculation of the number of subjects needed in this kind of pilot study. Low numbers have provided good results in other dry powder inhalation studies of antimicrobials, so we think that twelve participants are sufficient to make a good impression of safety and tolerability [37, 47–49]. Another limitation is that we gave single doses while in a clinical setting one will have to administer multiple doses for treatment of SARS-CoV-2 infection. However, single ascending doses are not uncommon in phase 1 clinical studies on safety and tolerability. Next to that, we believe that HCQ dry powder can be used safely for multiple times and in asthma patients as well, since aerosolized HCQ has been applied in a phase I clinical study to assess safety for use in asthma and was concluded to be safe and well tolerated in 31 healthy individuals in doses up to 20 mg daily for 7 days [34]. In 2006, a phase 2 clinical trial with aerosolized HCQ as anti-inflammatory treatment for patients with asthma showed that a dose of 20 mg daily was tolerated for up to 21 days. However, the development was stopped as it failed to meet the primary clinical endpoints for effective asthma treatment; relative improvement in $FEV_1$ compared to baseline was not statistically significant after treatment compared to placebo. None of these participants had significant ECG changes and side effects consisted of headache and nausea only [34].

So far, only *in vitro* experiments on Vero cells have shown efficacy and data from human pulmonary concentrations and thus local efficacy data are lacking [9, 10]. That is a limitation in this study as well since we only measured systemic HCQ concentrations. Ideally this should be measured locally, in lung tissue, bronchoalveolar lavage and epithelial lining fluid, which is to be done in a phase 1b study with bronchoscopy.

HCQ serum concentrations were below the quantification limit of 10 μg/L in all participants and irrespective of dose or timepoints. Possibly our timing of blood sampling was not

optimal. If the maximum concentration ($C_{max}$) occurs very shortly after inhalation ($T_{max}$), we might have missed this peak concentration with our first blood sample drawn after 30 minutes. This is supported by the only other available pharmacokinetic data from a phase I clinical trial with aerosolized HCQ that came available after our study protocol was developed. Fifteen participants inhaled single doses of 5, 10 or 20 mg HCQ. Reported HCQ serum concentrations were mean $C_{max}$ between 22 and 69 μg/L with an early $T_{max}$ within 2–3 minutes. The reported systemic exposure was very low (7–54 μg*h/L), suggesting distribution within 30 minutes [34]. Local lung concentrations are expected to be higher after inhalation of HCQ compared to oral administration, even though the highest dose of inhaled HCQ in this study (20 mg) is only a small fraction of the usual oral HCQ dose (200–800 mg). For example, if the delivered dose of 15 mg homogeneously distributes over the lung tissue (843 mL) [50], a lung tissue HCQ concentration of approximately 40 μM could be achieved. Preliminary results from our own experiments in primary human epithelial cells indicate that HCQ concentrations of approximately 20 to 40 μM do result in a significant reduction in viral load after SARS-CoV-2 infection (manuscript in preparation). This contrasts to the lack of antiviral effect found by Mulay *et al.* at 10 μM HCQ, which may be explained by the fourfold lower concentration [51]. Based on these considerations, effective concentrations can potentially be achieved in lung tissue after inhalation of 20 mg HCQ. These high HCQ concentrations and the superior potential of inhaled over oral HCQ should be the new starting point of any further clinical studies on the activity of HCQ against SARS-CoV-2.

In conclusion, single HCQ inhalations up to 20 mg using the Cyclops are safe and generally well tolerated by the participants of this study, except for minor cough and bitter taste. These positive results and the superior safety and efficacy potential of inhaled over oral HCQ strongly encourage the execution of further clinical studies with inhaled HCQ to battle this COVID-19 pandemic.

## Supporting information

**S1 Checklist. CONSORT checklist.**
(DOC)

## Acknowledgments

We thank Marieke van Rossum and Marina Zenina for their help in the execution of this study.

## Author Contributions

**Conceptualization:** P. Hagedoorn, M. G. G. Sturkenboom, F. Grasmeijer, M. S. Bolhuis, I. Sibum, H. A. M. Kerstjens, H. W. Frijlink, O. W. Akkerman.

**Formal analysis:** Y. A. de Reus, H. A. M. Kerstjens, H. W. Frijlink, O. W. Akkerman.

**Investigation:** Y. A. de Reus, P. Hagedoorn, M. G. G. Sturkenboom, O. W. Akkerman.

**Methodology:** P. Hagedoorn, M. G. G. Sturkenboom, F. Grasmeijer.

**Project administration:** Y. A. de Reus.

**Supervision:** H. A. M. Kerstjens, O. W. Akkerman.

**Writing – original draft:** Y. A. de Reus, O. W. Akkerman.

**Writing – review & editing:** P. Hagedoorn, M. G. G. Sturkenboom, F. Grasmeijer, M. S. Bolhuis, I. Sibum, H. A. M. Kerstjens, H. W. Frijlink.

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
