## [Decision Letter · Decision Letter 0]

19 Mar 2021

PONE-D-20-37157

Tolerability and Pharmacokinetic Evaluation of Inhaled Dry Powder Hydroxychloroquine in Healthy Volunteers

PLOS ONE

Dear Dr. Akkerman,

Thank you for submitting your manuscript to PLOS ONE. After careful consideration, we feel that it has merit but does not fully meet PLOS ONE’s publication criteria as it currently stands. Therefore, we invite you to submit a revised version of the manuscript that addresses the points raised during the review process.

Five experts in the field reviewed your manuscript. Please ensure that you thoroughly respond to all of their comments.

We look forward to receiving your revised manuscript.

Kind regards,

Susan Hepp

Staff Editor

PLOS ONE

Journal Requirements:

2. Please provide the full name of the Institutional Review Board that approved your study.

3. In your Methods section, please provide additional information about the participant recruitment method and the demographic details of your participants. Please ensure you have provided sufficient details to replicate the analyses such as:

a) the recruitment date range (month and year),

b) a statement as to whether your sample can be considered representative of a larger population, and

c) a description of how participants were recruited.

4. Please include your tables as part of your main manuscript and remove the individual files. Please note that supplementary tables (should remain/ be uploaded) as separate "supporting information" files

7. Thank you for stating the following in the Competing Interests section:

'I have read the journal's policy and the authors of this manuscript have the following competing interests: The employer of Paul Hagedoorn has a royalty agreement with AstraZeneca on the sales of the Genuair Inhaler; In addition, Paul Hagedoorn has a patent WO2015NL50413 20150605 licensed.

Floris Grasmeijer is employed by PureIMS, the manufacturer of the Cyclops inhaler.

The employer of Erik Frijlink has a license agreement with PureIMS on the Cyclops patent (WO2015/187025).'

We note that one or more of the authors are employed by a commercial company:PureIMS.

8. We note that you have a patent relating to material pertinent to this article. Please provide an amended statement of Competing Interests to declare this patent (with details including name and number), along with any other relevant declarations relating to employment, consultancy, patents, products in development or modified products etc. Please confirm that this does not alter your adherence to all PLOS ONE policies on sharing data and materials, as detailed online in our guide for authors http://journals.plos.org/plosone/s/competing-interests by including the following statement: "This does not alter our adherence to  PLOS ONE policies on sharing data and materials.” If there are restrictions on sharing of data and/or materials, please state these. Please note that we cannot proceed with consideration of your article until this information has been declared.

Additional Editor Comments (if provided):

Reviewers' comments:

Reviewer's Responses to Questions

**Comments to the Author**

1. Is the manuscript technically sound, and do the data support the conclusions?

Reviewer #1: Yes

Reviewer #2: Yes

Reviewer #3: Yes

Reviewer #4: Partly

Reviewer #5: Yes

2. Has the statistical analysis been performed appropriately and rigorously? 

Reviewer #1: N/A

Reviewer #2: Yes

Reviewer #3: Yes

Reviewer #4: N/A

Reviewer #5: I Don't Know

3. Have the authors made all data underlying the findings in their manuscript fully available?

Reviewer #1: Yes

Reviewer #2: No

Reviewer #3: Yes

Reviewer #4: No

Reviewer #5: Yes

4. Is the manuscript presented in an intelligible fashion and written in standard English?

Reviewer #1: Yes

Reviewer #2: Yes

Reviewer #3: Yes

Reviewer #4: Yes

Reviewer #5: Yes

5. Review Comments to the Author

Reviewer #1: This is a well written concise manuscript describing a small safety study in healthy volunteers to explore the use of inhaled dry powder hydroxychloroquine to treat COVID 19 in the future. The authors correctly point out that this study does not provide proof of principle that the dose and administration used here produces the drug levels in alveolar tissue that would definitelyinhibit viral invasion. The authors could add some sentences to discuss whether a separate study to attempt to measure alveolar levels in healthy subjects should be done, I think the authors should comment on whether this inhalerwould produce alveolar deposition or whether it is limited to larger airways.A comment on ANY DEPOSITION STUDIES would help

Reviewer #2: The major problem with the study is the sample size which is 12. However, due to the fact that the authors use only a t-test, this is potentially acceptable.

The authors should add a sample size estimation based on reasonable assumptions for the effect size to demonstrate that for the studied problem this small sample size is tolerable.

In addition, please emphasize in the discussion this issue.

Reviewer #3: This manuscript is a well written paper with clear study design and conclusions. However, there are some minor points to be addressed.

1. Please describe more details for the study design including dosing sequence, wash-out period, food (fasting or fed), restrictions.

2. Please describe or discuss the rationale for the time points of FEV1 assessments and ECG assessment.

3. It seems to be better to present the adverse events according to dose in Table 3.

4. This study aimed to evaluate the tolerability and PK after a single dose. However, in clinical setting, multiple doses will be needed for the treatment of COVID-19. Please add the limitation of this single dose study in Discussion.

Reviewer #4: This paper describes a simple but useful trial to test, in healthy volunteers, the pulmonary safety and tolerability of an inhaled, dry-powder formulation of hydroxychloroquine. It deserves rapid publication, because it's of topical interest as regards the treatment of COVID-19.

The authors should add comments or discussion about the following weaknesses of the trial:

1) The trial is open-label and has no placebo control, so no statistical hypothesis testing is justified. Also, the lack of placebo greatly weakens the assessment of tolerability and safety, including the main pulmonary outcome measure (FEV1).

2) the trial did not include any measures of pulmonary inflammation (such as exhaled NO) or gas permeability (eg CO transfer factor)

3) In line 235, the authors refer to the rapid "elimination" of hydroxychloroquine: the word "distribution" would be more accurate.

4) There are a few typographical errors to be corrected. Also, the phrase "allow for" has been used throughout: the "for" is incorrect in this context, and "allow" should be used on its own.

5) Complete data are not available

Reviewer #5: The article is well written and can have an impact in clinical practice. The major disadvantage is related to the time when the blood samples were taken, too late to measure the plasma concentration of the drug.

Table 3 mention "respiratory infection", but this term does not appear in the manuscript. How did you evaluate this side effect?

In my opinion, "inhaled dry powder HCQ is safe" is too enthusiastic statement, because you have followed a small number of patients for a short time after single increasing doses of 5, 10 and 20 mg. I think you need to mention this within the limits of the study.

6. PLOS authors have the option to publish the peer review history of their article (what does this mean?). If published, this will include your full peer review and any attached files.

Reviewer #1: **Yes: **Diana Bilton

Reviewer #2: No

Reviewer #3: No

Reviewer #4: **Yes: **Steve Warrington

Reviewer #5: No

---

## [Author Response · Author response to Decision Letter 0]

26 Apr 2021

We would like to thank the editor and reviewers for their time, critical view and suggestions to improve our manuscript. 

The following additional requirements were implemented:

1. We applied the PLOS ONE style requirements

2. We provided the full name of the Institutional Review board [line 116].

3. A and C: Additional information about the participant recruitment method, recruitment date range [line 109-112] and table with demographic details of the participants [table 2] are added. 

B: We made a statement in the discussion section whether our sample can be considered representative of a larger population [ see comment on reviewer 2]

4. We included the tables with separate captions in the manuscript and removed them as separate files. 

5. The figure was given a separate caption in the manuscript. 

6. We did not add new supporting information files with this resubmission

7. We updated the Competing Interests Statement and the Funding Statement as mentioned in cover letter and in the manuscript. 

8. We declared the use of a patented inhaler called the Cyclops with number WO2015/187025 in the cover letter. 

9. We reviewed and completed the reference list. The references to the unpublished data of Dayton et al were replaced [reference 37, 38, 39], since their results were not directly available to us and only published after personal communication in the article of Kavanagh et al [reference 34]. The references belonging to the retracted article of Mehra et al [56] and the retraction notice by Watson et al [57] are removed. New references are added as mentioned in the response to the reviewers.

Below is our response to each point raised by the reviewers. Line numbers correspond with the lines in the ‘revised mancuscript track and changes’

Reviewer 1: 

This is a well written concise manuscript describing a small safety study in healthy volunteers to explore the use of inhaled dry powder hydroxychloroquine to treat COVID 19 in the future. The authors correctly point out that this study does not provide proof of principle that the dose and administration used here produces the drug levels in alveolar tissue that would definitely inhibit viral invasion. 

The authors could add some sentences to discuss whether a separate study to attempt to measure alveolar levels in healthy subjects should be done.

I think the authors should comment on whether this inhaler would produce alveolar deposition or whether it is limited to larger airways. A comment on ANY DEPOSITION STUDIES would help

Thank you for these suggestions. 

It is correct that this study does not provide proof of principle that the dose and administration used here produces alveolar tissue concentrations that inhibit viral invasion or replication. The limitation of the study is that we only measured hydroxychloroquine systemically – in blood samples – but ideally this should be measured in lung tissue, BAL, and epithelial lining fluid as well. It would be of great interest to know what concentration can be achieved after inhalation and compare this with oral dosing. We added this limitation to the discussion section [line 290-294]. 

We made the expectation that there will be alveolar deposition with use of the Cyclops more clear in the introduction section by adding information about the particle size produced by the inhaler [line 92-97].

Reviewer 2: 

The major problem with the study is the sample size which is 12. However, due to the fact that the authors use only a t-test, this is potentially acceptable. 

The authors should add a sample size estimation based on reasonable assumptions for the effect size to demonstrate that for the studied problem this small sample size is tolerable. In addition, please emphasize in the discussion this issue.

We agree on the comment of the small sample size. It is difficult to make a specific calculation of the number of subjects needed in this pilot study. Low numbers have provided good results in other dry powder inhalation studies of antibiotics [41-44], so we think that 12 participants are sufficient to make a good impression of safety and tolerability. This is supported specifically for HCQ by the phase 1 study of aerosolized HCQ in 31 healthy individuals and the phase 2 with aerosolized HCQ in patients with asthma [34]. We added a note of this in the discussion section and provided new supporting references accordingly [line 271-287]. 

Reviewer 3: 

This manuscript is a well written paper with clear study design and conclusions. However, there are some minor points to be addressed.

1. Please describe more details for the study design including dosing sequence, wash-out period, food (fasting or fed), restrictions.

We provided extra information in the manuscript regarding the wash-out period of at least 4 days in the methods section. As mentioned in the manuscript the dosing sequence consisted of single doses, starting with a dose of 5 mg, ascending tot 10 and finally 20mg [line 112-114]. Feeding state and restrictions are important point to address in drug research, but since the drug in this study is deposited and resorbed from the lungs it holds less relevance. Therefore we did not document feeding state or other restrictions in the manuscript. 

2. Please describe or discuss the rationale for the time points of FEV1 assessments and ECG assessment.

The timepoints for spirometry were chosen shortly after inhalation, because bronchoconstriction as a side effect is expected to occur shortly after inhalation due to direct irritating effects. This has been shown in other studies with inhaled antimicrobials, for example in the article of Hoppentocht et al where drop in FEV1 > 10% is observed within 95 minutes after inhalation of dry powder tobramycin [41][line 135-138]. 

The risk of QTc prolongation increases with higher systemic HCQ concentrations. We added ECG assessment as a safety parameter, although (high) systemic concentrations were not to expected after local administration in a dose that is just a fraction of the oral dose [line 260-268]. Since there was no data available about the absorption rate of HCQ from the lungs we based our time point on the time that HCQ reaches a maximum concentration after oral dosing, which is within 2-4.5 hours. For logistical reasons we planned it at the end of the study day which was 3,5 hours after inhalation [line 295-296]. 

3. It seems to be better to present the adverse events according to dose in Table 3.

Thank you for this suggestion. We updated the table en presented adverse events according to dose and in absolute numbers instead of percentages in table 3. 

4. This study aimed to evaluate the tolerability and PK after a single dose. However, in clinical setting, multiple doses will be needed for the treatment of COVID-19. Please add the limitation of this single dose study in Discussion.

We agree that the single dose is a limitation in our study since in COVID-19 treatment multiple doses will be needed. We added this to the limitations in the discussion section [lines 275-277].

Reviewer 4:

This paper describes a simple but useful trial to test, in healthy volunteers, the pulmonary safety and tolerability of an inhaled, dry-powder formulation of hydroxychloroquine. It deserves rapid publication, because it's of topical interest as regards the treatment of COVID-19. The authors should add comments or discussion about the following weaknesses of the trial:

1. The trial is open-label and has no placebo control, so no statistical hypothesis testing is justified. The lack of placebo weakens the assessment of tolerability and safety, including the main pulmonary outcome measure. (FEV1).

The study indeed is open label without placebo control. We believe this limitation is more important when one does find safety or tolerability problems and perspective is needed, but not so much when problems found are minor anyway.

2. The trial did not include any measures of pulmonary inflammation (such as exhaled NO) or gas permeability (eg CO transfer factor)

We agree that pulmonary inflammation as such was not measured, nor potential proxies such as FeNO or KCO. This will be interesting to consider in a next, phase 1b study in which we will aim for measuring direct local concentrations in the airways and BAL.

3. In line 235, the authors refer to the rapid "elimination" of hydroxychloroquine: the word "distribution" would be more accurate.

Indeed the word distribution is more accurate, we replaced the word elimination. 

4. There are a few typographical errors to be corrected. Also, the phrase "allow for" has been used throughout: the "for" is incorrect in this context, and "allow" should be used on its own.

We deleted the word ‘for’ in this context and checked the manuscript again for typographical errors. 

5. Complete data are not available

All relevant data are within the paper. The complete data set can be requested from the authors.

Reviewer 5: 

The article is well written and can have an impact in clinical practice. The major disadvantage is related to the time when the blood samples were taken, too late to measure the plasma concentration of the drug.

We agree this is an disadvantage and added this as a limitation of the study in the discussion section [line 293-299]. Timepoints for drawing the blood samples were based on the rate of absorption after oral administration of HCQ, since data after pulmonary absorption were lacking at time of writing of the study protocol. Based on results and current knowledge blood samples should be taken at earlier time points as well in future studies. 

Table 3 mention "respiratory infection", but this term does not appear in the manuscript. How did you evaluate this side effect?

In table 3 we changed ‘respiratory infection/sore throat’ to ‘sore throat’. Participants complaining of sore threat or other upper respiratory tract infection symptoms were provided a swab for respiratory viruses of which one tested positive for SARS-CoV-2, as mentioned in the manuscript [line 191-194]. 

In my opinion, "inhaled dry powder HCQ is safe" is too enthusiastic statement, because you have followed a small number of patients for a short time after single increasing doses of 5, 10 and 20 mg. I think you need to mention this within the limits of the study.

We agree on this comment and added this as a limitation in the discussion section. See comments made by reviewer 2 and reviewer 3. In addition we nuanced the conclusion [line 45-47 and line 334-335].

---

## [Decision Letter · Decision Letter 1]

22 Dec 2021

PONE-D-20-37157R1Tolerability and Pharmacokinetic Evaluation of Inhaled Dry Powder Hydroxychloroquine in Healthy VolunteersPLOS ONE

Dear Dr. Akkerman,

Thank you for submitting your manuscript to PLOS ONE. After careful consideration, we feel that it has merit but does not fully meet PLOS ONE’s publication criteria as it currently stands. Therefore, we invite you to submit a revised version of the manuscript that addresses the points raised during the review process.

We look forward to receiving your revised manuscript.

Kind regards,

Kaisar Raza

Academic Editor

PLOS ONE

Journal Requirements:

Reviewers' comments:

Reviewer's Responses to Questions

**Comments to the Author**

1. If the authors have adequately addressed your comments raised in a previous round of review and you feel that this manuscript is now acceptable for publication, you may indicate that here to bypass the “Comments to the Author” section, enter your conflict of interest statement in the “Confidential to Editor” section, and submit your "Accept" recommendation.

Reviewer #3: All comments have been addressed

Reviewer #6: (No Response)

2. Is the manuscript technically sound, and do the data support the conclusions?

Reviewer #3: Yes

Reviewer #6: Yes

3. Has the statistical analysis been performed appropriately and rigorously? 

Reviewer #3: Yes

Reviewer #6: Yes

4. Have the authors made all data underlying the findings in their manuscript fully available?

Reviewer #3: Yes

Reviewer #6: Yes

5. Is the manuscript presented in an intelligible fashion and written in standard English?

Reviewer #3: Yes

Reviewer #6: Yes

6. Review Comments to the Author

Reviewer #3: All comments and suggestions are properly reflected and addressed. So, it is acceptable for a new publication.

Reviewer #6: The study design was clear, and the manuscript was well written in general. I would recommend authors to briefly mention the method of development of HCQ dry powder formulation and it’s aerosolization behavior in vitro as this study couldn’t test the lung concentrations of drug before acceptance for publication.

7. PLOS authors have the option to publish the peer review history of their article (what does this mean?). If published, this will include your full peer review and any attached files.

Reviewer #3: No

Reviewer #6: No

---

## [Author Response · Author response to Decision Letter 1]

4 Jan 2022

We would like to thank the editor and reviewers for their time, critical view and suggestions to improve our manuscript. 

The following additional requirements were implemented based on the suggestion from reviewer 6. The study design was clear, and the manuscript was well written in general. I would recommend authors to briefly mention the method of development of HCQ dry powder formulation and it’s aerosolization behavior in vitro as this study couldn’t test the lung concentrations of drug before acceptance for publication

In response to reviewer 6 we added additional information to the methods section, under 'study drug', about the development of the HCQ dry powder formulation and it’s behavior in vitro. [line 107-115].

The HCQ Cyclops was produced by PureIMS B.V. (Roden, the Netherlands). For this, hydroxychloroquine sulphate was comicronized by air jet milling with 4% L-leucine to improve formulation dispersion. Each Cyclops contained a nominal dose of 5 mg or 10 mg HCQ; the 20 mg dose was administered as 2 successive inhalations with 10 mg of HCQ. The doses were comparable with known doses of inhaled nebulized HCQ and thus expected to be safe (34, 36). Furthermore, to enhance dose emission from the inhaler, 5 mg of coarse lactose was added to the dose compartment. At 4 kPa, 85% of the nominal dose was emitted from Cyclops with a fine particle fraction < 5 µm of 74% (i.e. 63% of the nominal dose), as determined by laser diffraction analysis.

---

## [Decision Letter · Decision Letter 2]

18 May 2022

PONE-D-20-37157R2Tolerability and Pharmacokinetic Evaluation of Inhaled Dry Powder Hydroxychloroquine in Healthy VolunteersPLOS ONE

Dear Dr. Akkerman,

Thank you for submitting your manuscript to PLOS ONE. After careful consideration, we feel that it has merit but does not fully meet PLOS ONE’s publication criteria as it currently stands. Therefore, we invite you to submit a revised version of the manuscript that addresses the points raised during the review process.

 Please revise on the following grounds too:

This is a dose-escalation Phase I repeated-measures trial. The authors studied the safety and tolerability of dry HCQ through inhalation with dose escalation among the healthy volunteers. The tolerability was assessed by the measures on PFT and ECG and the safety through the adverse events. The design of the study seems appropriate and well-written manuscript. However, the authors need to make more changes to shape the manuscript nicely. Below are some comments: 1. The topic is misleading on 'Pharmacokinetics Evaluation. I have not seen any in the manuscript related to that. 2. The statement in the first two paragraphs of the introduction section is obsolete. The authors need to update at least as of December 31, 2021. 3. The authors must mention many available treatment modalities, including the recent EUA COVID pill. 4. Why was the oral HCQ not advised by FDA and EMA? The authors need to refer to the RECOVERY Trial and the prospective European DISCOVERY and WHO SOLIDARITY trials. Those details would enhance the manuscript. 5. The statistical analytic plans are critically important in ensuring appropriate reporting of clinical trials. The significance reported using the methods seems irrelevant due to the small sample size and the authors' need to consider using the exact tests. The main focus is the safety and tolerability, not the statistical significance of the measures collected. 6. There is no uniformity in using the terminology. In some places in the manuscript, the authors use 'volunteer' and state 'participant' in many areas. 7. The second paragraph in the results section (lines 165- 184) should be converted into a table by listing all event names with a number of participants. Then, delete the entire text from that section. 8. Not clear, the authors screened 12 participants, and all are eligible and enrolled in the study? If it is more than that, it has to be reported. 9. The statements in the discussion section (Lines 217-227) should highlight the importance of the study and focus on elaborating the contents based on the data with supporting evidence (any prior cited research). The statistical values reported in that paragraph in the lines 229-240 should be deleted. 10. The authors described why the timing was chosen (Lines 268-272) for blood collection. These statements should move to the methods section. 11. There is not enough in the discussion to claim this study is warranted. A strong justification is still missing: why is the clinical research warranted? How would the proposed method help in the clinical arena? 12. Line 198: Table 1: Check the spelling for 'Cough.' It says 'Couch.' 13. Line 203: Table 3: Last column 'Percentage of patients' is meaningless- should be excluded. Add the percentage under each dose column N (%). 14. Line 205: The change in value for 10 mg HCQ from baseline to 95 minutes post inhalation is reported as 0.21. All the other values are less than zero. The authors need to double-check the calculation of this declared value.

We look forward to receiving your revised manuscript.

Kind regards,

Kaisar Raza

Academic Editor

PLOS ONE

Journal Requirements:

Reviewers' comments:

Reviewer's Responses to Questions

**Comments to the Author**

1. If the authors have adequately addressed your comments raised in a previous round of review and you feel that this manuscript is now acceptable for publication, you may indicate that here to bypass the “Comments to the Author” section, enter your conflict of interest statement in the “Confidential to Editor” section, and submit your "Accept" recommendation.

Reviewer #6: All comments have been addressed

2. Is the manuscript technically sound, and do the data support the conclusions?

Reviewer #6: Yes

3. Has the statistical analysis been performed appropriately and rigorously? 

Reviewer #6: Yes

4. Have the authors made all data underlying the findings in their manuscript fully available?

Reviewer #6: Yes

5. Is the manuscript presented in an intelligible fashion and written in standard English?

Reviewer #6: Yes

6. Review Comments to the Author

Reviewer #6: I would appreciate authors for addressing all the comments. Now, I would recommend accepting this manuscript for publication.

7. PLOS authors have the option to publish the peer review history of their article (what does this mean?). If published, this will include your full peer review and any attached files.

Reviewer #6: No

---

## [Author Response · Author response to Decision Letter 2]

1 Jul 2022

We would like to thank the editor and reviewer for their time, critical view and suggestions to improve our manuscript. 

1. The topic is misleading on 'Pharmacokinetics Evaluation. I have not seen any in the manuscript related to that. 

The pharmacokinetic evaluation consisted of measuring serum HCQ concentrations predose, 30 minutes, 2 and 3,5 hours after inhalation. Serum HCQ concentrations are reported in the manuscript. Please refer to line 202-206

“In all participants, the serum HCQ concentrations sampled predose, 30 minutes, 2 and 3.5 hours, were below the detection level of 10 µg/L, irrespective of dose. The mean delivered dose was 3.16 mg (63%) after administration of 5 mg HCQ, 6.95 mg (70%) after administration of 10 mg HCQ and 14.86 mg (75%) after administration of 20 mg HCQ. Based on the recorded inspiratory flow parameters, all participants correctly performed the inhalation maneuvers.”

2. The statement in the first two paragraphs of the introduction section is obsolete. The authors need to update at least as of December 31, 2021.

Thank you, we agree. Unfortunately, the manuscript has been under review for two very long periods at the journal. The information is now updated. 

3. The authors must mention many available treatment modalities, including the recent EUA COVID pill. 

Thank you for this suggestion. We added a paragraph with a short summary of current treatment advice according to the WHO living guideline accessed June 26th, 2022. 

4. Why was the oral HCQ not advised by FDA and EMA? The authors need to refer to the RECOVERY Trial and the prospective European DISCOVERY and WHO SOLIDARITY trials. Those details would enhance the manuscript. 

Yes, it is important to know that oral HCQ was not adviced because of a lack of effect on mortality. In line 78-81 we now refer to this decision and the mentioned trials. 

5. The statistical analytic plans are critically important in ensuring appropriate reporting of clinical trials. The significance reported using the methods seems irrelevant due to the small sample size and the authors' need to consider using the exact tests. The main focus is the safety and tolerability, not the statistical significance of the measures collected.

We would like to thank the reviewer for this advice. Since the main focus is indeed the safety and tolerability and the sample size is small we decided not to mention the statistical analysis anymore and only use descriptive statistics. Therefore we changed the text in the data management and statistics section and removed the p-values from the tables. 

6. There is no uniformity in using the terminology. In some places in the manuscript, the authors use 'volunteer' and state 'participant' in many areas. 

Thank you for this suggestion to make the manuscript more uniform. As volunteers participate in the study they can also be referred to as participants. The term healthy volunteer is now only used in the methods sections, in the rest of the manuscript we use the term participant. 

7. The second paragraph in the results section (lines 165- 184) should be converted into a table by listing all event names with a number of participants. Then, delete the entire text from that section. 

Thank you, we agree that a table gives the best overview of the events, they are mentioned in table 3. To our opinion, the text adds more detailed information to this. Therefore we decided not to remove the text. 

8. Not clear, the authors screened 12 participants, and all are eligible and enrolled in the study? If it is more than that, it has to be reported. 

All participants screened were eligible and were enrolled in the study. This has now been made more clear in the text.

9. The statements in the discussion section (Lines 217-227) should highlight the importance of the study and focus on elaborating the contents based on the data with supporting evidence (any prior cited research). The statistical values reported in that paragraph in the lines 229-240 should be deleted. 

Thank you, we tried to make the importance of the study more clear in the first paragraph of the discussion and added citations to previous mentioned research in the first and second paragraph as well. 

10. The authors described why the timing was chosen (Lines 268-272) for blood collection. These statements should move to the methods section. 

We moved the statement to the methods section as suggested. 

11. There is not enough in the discussion to claim this study is warranted. A strong justification is still missing: why is the clinical research warranted? How would the proposed method help in the clinical arena? 

There is not a good explanation why hydroxychloroquine lacks a clinical effect in the treatment of COVID-19 despite the promising results of in vitro studies and the assumed mechanism of action. We tried to state this more clear in the first section of the discussion section, addressing that we hypothesize that high enough concentrations are not reached in the lung where needed when dosed systemically, because of limitations of systemic toxicity, but might be reached by inhaled dosing. 

12. Line 198: Table 1: Check the spelling for 'Cough.' It says 'Couch.' 

We corrected the spelling

12. Line 203: Table 3: Last column 'Percentage of patients' is meaningless- should be excluded. Add the percentage under each dose column N (%). 

Thank you. We excluded the last column and added the percentages under the remaining column as suggested. 

14. Line 205: The change in value for 10 mg HCQ from baseline to 95 minutes post inhalation is reported as 0.21. All the other values are less than zero. The authors need to double-check the calculation of this declared value.

We checked this value and the value of +0.21 % is correctly documented.

---

## [Editor Report · Decision Letter 3]

13 Jul 2022

Tolerability and Pharmacokinetic Evaluation of Inhaled Dry Powder Hydroxychloroquine in Healthy Volunteers

PONE-D-20-37157R3

Dear Dr. Akkerman,

We’re pleased to inform you that your manuscript has been judged scientifically suitable for publication and will be formally accepted for publication once it meets all outstanding technical requirements.

Kind regards,

Kaisar Raza

Academic Editor

PLOS ONE
---

## [Editor Report · Acceptance letter]

26 Jul 2022

PONE-D-20-37157R3 

Tolerability and Pharmacokinetic Evaluation of Inhaled Dry Powder Hydroxychloroquine in Healthy Volunteers 

Dear Dr. Akkerman:

I'm pleased to inform you that your manuscript has been deemed suitable for publication in PLOS ONE. Congratulations! Your manuscript is now with our production department. 

Kind regards, 

on behalf of

Dr. Kaisar Raza 

Academic Editor

PLOS ONE